# Older People Living Alone: A Predictive Model of Fall Risk

**DOI:** 10.3390/ijerph20136284

**Published:** 2023-07-03

**Authors:** Isabel Lage, Fátima Braga, Manuela Almendra, Filipe Meneses, Laetitia Teixeira, Odete Araújo

**Affiliations:** 1School of Nursing, University of Minho, 4710-057 Braga, Portugal; ilage@ese.uminho.pt (I.L.); fbraga@ese.uminho.pt (F.B.);; 2Nursing Research Centre, University of Minho, 4710-057 Braga, Portugal; 3School of Engineering, University of Minho, 4710-057 Braga, Portugal; 4Centro de Computação Gráfica, 4800-058 Guimarães, Portugal; 5Algoritmi Research Centre, University of Minho, 4710-057 Braga, Portugal; 6ICBAS, University of Porto, 4050-313 Porto, Portugal; 7CINTESIS@RISE, ICBAS, University of Porto, 4050-313 Porto, Portugal; 8Health Sciences Research Unit: Nursing (UICISA:E), Nursing School of Coimbra (ESEnfC), 3045-043 Coimbra, Portugal

**Keywords:** older people, living alone, falls prediction, risk of falling, nursing

## Abstract

Falls in older people are a result of a combination of multiple risk factors. There are few studies involving predictive models in a community context. The aim of this study was to determine the validation of a new model for predicting fall risk in older adults (65+) living alone in community dwellings (n = 186; n = 117) with a test–retest reliability study. We consider in the predictive model the significant factors emerged from the bivariate analysis: age, zone, social community resources, physical exercise, self-perception of health, difficulty to keep standing, difficulty to sit and get up from a chair, strain to see, use of technical devices, hypertension and number of medications. The final model explained 28.5% of the risk of falling in older adults living alone in community dwellings. The AUC = 0.660 (se = 0.065, IC 95% 0.533–0.787, *p* = 0.017). The predictive model developed revealed a satisfactory discriminatory performance of the model and can contribute to clinical practice, with respect to the evaluation of risk of falling in this frailty group and preventing falls.

## 1. Introduction

Evidence has shown that the world’s population is aging at an alarming rate. By 2050, the world’s population of people aged 60 years and older will double (2.1 billion). The number of persons aged 80 years or older is expected to triple between 2020 and 2050 to reach 426 million. Approximately, two-thirds of the world’s population over 60 years will live in low- and middle-income countries [1].

In the European Union by the year 2050, such as in Portugal, the population is set to become much older, increasing the risk of frailty in older adults as a consequence of physical, mental and social problems which compromise successful ageing. In 2021, there were 2,424,122 older people aged 65 and over living in Portugal [2]. According to Eurostat, in 2015, more than 24% of older Portuguese people were living alone at home [3].

As a consequence of the ageing population phenomena and age-related frailty and diseases, one of the major concerns is the higher risk of falling [4,5]. Worldwide, falls are considered the leading cause of injury-related deaths and non-fatal injuries for adults in general but particularly among older people [6,7,8]. One in every three persons over 65 years old and one in every two persons over the age of 85 years old will experience a fall at least once a year [4,5,6]. In many situations, falls are the most serious and frequent home accident among older people, leading to devastating consequences, i.e., causing a loss of independence, a higher level of morbidity and isolation and leading to a major risk of long-care needs, hospitalization, having to move into a nursing or residential home and, in other cases, to early death [8,9,10]. In addition to the direct human costs, the healthcare costs are staggering, with falls estimated to cost EUR 25B a year in the EU. In the US, the direct costs of falls account for 6% of total Medicare spending. Evidence has also shown that falls can be preventable, making it increasingly a priority to create ecosystems involving health and social organizations to aware the population in general, and the older population in particular to prevent falls [11,12]. According to the National Institute for Health and Care Excellence [5,13] in the UK, strength and balance exercises should be prescribed to facilitate the prevention of falls. A meta-analysis conducted by Deandrea et al. [13] confirmed the multifactorial etiology and considered 31 risk factors regarding the risk of falling, including sociodemographic, mobility, balance, sensory, psychologic, pathologies, medication use and other social variables, such as living alone. Living alone makes it more likely for an older individual to experience a fall [14,15], because living alone seems to be associated to lower social and emotional support, requiring more social and community health services as a consequence of functional decline, including technical aids or social workers [15].

Predictive risk models are suggested to identify the complicated relationship between modified and non-modified risk factors and how to identify and evaluate them. As a consequence of the existence of predictive models, more effective preventive measures, including personalized interventions to prevent falls in older people, are needed [16,17]. Evidence has shown that the major development and validation of predictive models for fall risk has occurred in the hospital environment [18,19]. Until now, there have been few studies involving predictive models in older people living in a community context. The advantages of these models can be huge, and identifying the older adults that are at high risk of falls is the first step in helping health care providers implement interventions to prevent falls and their consequences [20].

Nevertheless, in the community context, there are few reliable instruments that accurately predicts the risk of falling. There are difficulties at this level, because the existing instruments are focused on scenarios such as hospitals and institutions, and have a lack of sensitivity and specificity. In fact, they take time to be fulfilled, and not all professionals have the knowledge and skills necessary for the application of some functional and/or cognitive assessment tests [21,22]. Additionally, fall risk being multifactorial, there is no clear notion which factor can have the biggest weight on its determination in this community context. As such, in clinical practice, health professionals need a model that enables them to effectively assess the risk of falls in the community, which will allow them to implement comprehensive and sensitive interventions in older adults based on their individual needs [22], to associate the appropriate measures, for the right people, at the right time [20]. Thus, the objective of this study was to construct and validate a new model for predicting falls in older adults living alone in community dwellings.

## 2. Methods

### 2.1. Aims

The aim of this study was to assess the construction and validation of a new model for predicting fall risk in older adults living alone in community dwellings.

### 2.2. Design

This is a test–retest reliability study to predict falls in older adults living alone.

### 2.3. Participants

The population consisted of persons aged 65 years and older who live at home in the City of Braga (District). This project involved all older people who met the following eligibility criteria: (a) living alone; (b) without cognitive impairment; (c) ability to communicate.

### 2.4. Data Collection

Older people living alone in Braga District were enrolled in the study between 1 October 2018 and 30 April 2019. The participants were recruited from municipal autarchies and some daycare centers for older people. Additionally, the research team promoted initiatives and delivered sessions about falls prevention awareness to all older people, to sensitize them to this problem and to identify potential older people who were living alone and wanted to participate in this study. After each session regarding falls prevention, the research team invited all older people present who fulfill the eligibility criteria. In other situations, when older people did not attend the meetings, promoted by the research team, researchers (degree in nursing) in the Alertfalls project contacted potential participants by telephone. Once consent was obtained for a home visit, the team researchers fully informed older people about the content of the study: (i) purpose and objectives; (ii) anonymity; (iii) confidentiality; (iv) number of visits (2 visits in 15 days); (v) duration of the questionnaire application (30 min) and (vi) possibility of refusing to participate in the study. After signing the agreement, visits were scheduled according to each participant’s availability. All the procedures of data collection were harmonized with the team before questionnaire implementation. Participants who needed assistance to complete the questionnaire due to illiteracy or physical challenges (e.g., vision and/or hearing loss) received help from one of the researchers. Participants’ names were not included on the questionnaires to ensure confidentiality and an alphanumeric code was used in each questionnaire. 

A retest was performed 15 days after the first evaluation, considering the risk of drop out in older people (e.g., hospitalization/death). As stated in the literature, intervals should be spaced apart sufficiently to avoid fatigue, learning, or memory effects, yet be close enough to prevent genuine changes in the measured variable [23].

### 2.5. Context/Recruitment

Before the recruitment of the participants, the research team contacted parish councils to plan and aware older people for falls prevention, as well as the negative consequences of falls in this frail group living alone. It includes several awareness group sessions (average duration of 1 h) in accordance with the availability of the participants and, after each meeting, the team invited the participants (older people) who were living alone to participate in this study. Moreover, the participation of day centers was also included, because many older people are involved in activities organized by daycare centers. 

### 2.6. Variables/Measures

The variables included in the assessment protocol involved: (1) sociodemographic information (gender, age, zone, marital status, education level, retired, income per month); (2) social and health resources (informal caregiver, proximity informal network, health community resources, social community resources); (3) physical condition (physical exercise, self-perception of health); (4) functionality (bathing, dress/undress, ability to put on shoes/take off shoes, ability to eat and drink, taking care of own health, lack or presence of issues urination/defecation, preparing meals, ability to do housework, ability to leave home, use of public transport); (5) difficulties (hand, arms, legs, keep standing, sitting and getting up from a chair, staying seated, catching an object on the ground, standing up, walking, running, listening, seeing, talking, making efforts, memory, paying attention, orientation in time, orientation in space, sleep, use of public transport, understanding traffic signals); (6) technical devices (pacemaker, knee prosthesis, hip prosthesis, dental prosthesis, auditory orthotics, visual orthotics); (7) diagnoses and pain (hypertension, orthostatic hypotension, stroke, diabetes, osteoarticular disease, plantar changes, hearing impairment/vision, mental disorder, depression, permanent pain); (8) medication and management (number of medications, ability to buy the medicines, knowing the medicines, knowing the schedule/dose, preparing them correctly, conditioning them, knowing the indicators, knowing the adverse effects, knowing the drugs’ interaction, complying with the correct prescription, interrupting and/or suspending treatment by initiative, medication management support drug management); (9) risk behaviors (alcohol consumption, climbing up stairs, not paying attention to the environment, wearing poorly fitting shoes, wearing slippery slippers, wearing slippers of inappropriate size, not using product(s)/support devices); (10) physical environment/residential—internal (inadequate lighting, stairs, long distances between the different spaces and between the toilet and other spaces, confusing environment, loose carpets, loose electrical wires, presence of domestic animals, use of a walking stick, non-slip material in bathtub, grab bars in bath, side rails on both sides, furniture, adequate bed height, alarm system to be able to ask for help; (11) physical environment/residential—external (door sill, stairs, irregular floor, slippery floor, inadequate lighting).

### 2.7. Ethical Considerations

Permission to conduct the study was obtained from the *Conselho de Ética—Ciências da Vida e da Saúde* of the University of Minho (Reference number SECVS 033/2018). Prior to the data collection and after an explanation of the objectives of the study, all eligible participants returned informed written consent, and all ethical assumptions of confidentiality and voluntary participation were respected in accordance with the Declaration of Helsinki (59th amendment).

### 2.8. Data Analysis

Sample characteristics were obtained based on descriptive analysis. Association between fall and the measures included in the study protocol was performed, using a Chi-Square test and an independent sample *t*-test. A binary multivariable logistic regression model was performed to estimate the risk of falling, considering as predictive factors the significant factors obtained in the bivariate analysis. The evaluation of the discriminatory power of the risk model was based on the area under the curve (AUC) obtained from the receiver operator characteristic (ROC) curve. The predicted risk of falling was obtained by 1/(1 + exp(-LP)), where LP represents the linear predictor.

In all analysis, a significant level of 0.05 was considered.

### 2.9. Validity, Reliability and Rigour

Our study adopts a valid and tested cross-sectional survey design from research on older people. The measures of falls and all other factors evaluated have previously been used in studies among older people, which supported their predictive validity.

## 3. Results

### 3.1. Sample Characteristics

The sample (moment 1) comprised 186 older adults that were 65+ years old living alone. The overall mean age was 76.8 years (sd = 6.7 years), and most older adults were women (n = 144, 77.4%), widowed (n = 101, 54.3%), attended school between 1–4 years (n = 117, 62.9%), and lived in an urban zone (n = 148, 79.6%). The majority of older adults were retired (n = 184, 98.9%), with old-age pension as the main reason (n = 119, 65.4%), and received a median of 500 EUR/month (IQR = 350 EUR/month). In total, 161 participants (87.0%) had one or more children.

### 3.2. Predictive Model of Risk of Falling

Bivariate analysis was performed to identify potential predictive factors of falls, considering all aspects evaluated in the study protocol and described in the Methods section.

Based on these results, the variables considered in the predictive model, which are significant factors that emerged from the bivariate analysis, were age, zone, social community resources, physical exercise, self-perception of health, difficulty to keep standing, difficulty to sit and get up from the chair, difficulty to see, use technical devices, hypertension and number of medications. Additionally, gender was also considered in the final model.

The final model was presented in Table 1 and 28.5% of the risk of falling could be explained by the factors included in the model.

Despite only the variable “zone” remaining significant in the final model, with urban participants at three-fold higher odds of risk of falling compared with rural participants, the clinical significance of the other factors can be highlighted. 

Female and older participants present higher odds of risk of falling. Additionally, participants with social resources appear to have a small increase in risk, such as participants with difficulties to keep standing, sitting up and getting up from a chair and seeing. The use of technical devices, the presence of hypertension and the number of medications also appear to be associated with a greater tendency to the risk of falling. On the other hand, poor health and sedentarism seem to be protective factors of the risk of falling. The risk of falling for an older adult with specific characteristics could be obtained based on the following formula (considering 1 if the risk factor is present or 0 if absent):Linear predictor (LP) = −0.33 + (0.63* female) + (0.48* 75–84 years) + (0.36* 85+ years) + (1.16* urban) + (0.42* social community resources) + (−0.09* physical exercise) + (−1.09* poor health) + (−1.21* reasonable health) + (−1.56* good health) + (0.51* difficulty to keep standing) + 0.64* difficulty to sit and get up from the chair) + (0.21* difficulty to see) + (0.56* use of technical devices) + (0.71* Hypertension) + (0.74* 4–6 medications) + (0.38* 7 + medications)

#### Evaluation of Discriminatory Power of the Risk-Model

To evaluate the discriminatory capacity of the previous model, an ROC analysis was performed. Considering the sample 2, the probability was estimated based on the previous model, and results are presented in Figure 1. 

The AUC = 0.660 (se = 0.065, IC 95% 0.533–0.787, *p* = 0.017) revealed a satisfactory discriminatory performance of the model to identify older people with risk of falling.

### 3.3. Risk Profiles

Table 2 shows the applicability of the predictive model, considering three different profiles.

Profile 1 corresponds to the worst scenario, i.e., considering all characteristics related to high risk of falling. Profile 2 corresponds to profile 1, changing the characteristics that could be modified (social community resources, physical exercise, difficulties and technical devices). Finally, the last profile corresponds to profile 1 in terms of age, sex and self-perception of health, with all other factors modified (i.e., the changes made in profile 2 plus living in a rural area, without hypertension and a smaller number of medications).

Globally, profile 1 has a 99.5% risk of falling. When we changed the modified characteristics, this value decreases to 94.4% (Profile 2). Finally, when considering an opposite profile to the profile 1 (except for age, sex and self-perception of health), the risk decreases to 63.9%.

## 4. Discussion

This study revealed a satisfactory discriminatory performance of the model to quantify the risk of falling in older people living alone in community dwellings. At the moment, there are few studies involving predictive models in older people living in a community context in comparison with controlled scenarios, including hospital or nursing homes [18,19].

In this model, we introduced 12 variables obtained from community context data which were statistically relevant (bivariate analysis). The present model included the following factors: gender; age; zone; social community resources; physical exercise; self-perception of health; difficulty to keep standing; difficulty to sit and get up from a chair; difficulty to see; use of technical devices; hypertension; number of medications. The incidence of falls increases with age as a consequence of the physical and cognitive implications of advanced age [4,24,25]. Although the results show that women fall more, gender as a risk factor of falling is controversial [13]. Concerning the environment as a risk factor, there are few studies that can support the findings of our study. In practice, we believe that older people who are living in urban community contexts have a higher risk of falling because there are many obstacles on the pavement, and thus, they are exposed to a different risk every day. The scarcity of studies that compare both contexts in this population limits the comparison of results [26]. 

It is expected that the social community resources have a relevant role in helping people to minimize risks, but, interestingly, our results suggest that having social resources increase the risk of falling (OR 1.521). The reason can be associated with the type of support available or with the kind of limitations of the older people involved in this study. 

Exercise seems to be a protector of falling and decreases the probability of having a recurrent fall. Exercise programs that have shown to reduce falling risk primarily involve balance and functional exercises, while programs that might reduce falls include multiple exercise categories (typically balance and functional exercises plus resistance exercises) [27].

Regarding the factor associated with self-perception of health, although this indicator of health cannot be considered as a predictor of risk in many studies, in others ones, having “fair or poor” self-perception of health increases the risk for falling [28]. In our study, better self-perception of health increases the risk of falling. These results are very curious and, might show that “fair or poor” self-perception can be a protector. Although this variable is considered subjective, in the future, these findings need further research.

Loss of strength and balance leads to several difficulties in managing one’s activities of daily living, including self-care activities [29]. Furthermore, using technical devices, most of the time associated with physical decline, is also a significant risk factor, confirmed by the results (OR 1.746).

Pathologies have a meaningful impact as a predictor of falling. For example, having hypertension is associated with the risk of occurring falls [30], doubling the odds (OR 2.035), as well as the number of medications. From our results, we achieved that the intake between four and six medications also doubles/duplicates the risk of falls (OR 2.087). Recent literature supports our results [31,32].

Profiles 1, 2 and 3, presented in the results section, are linked to three scenarios: (1) profile 1, which is the worst scenario, showing 99.5% of risk of falling (including all variables of risk of falling); (2) profile 2, which introduces variables that can be modified (social community resources, physical exercise, difficulties and technical devices) and the risk of falling decreases to 94.4%; (3) when considering a profile inverse of profile 1 (except for age, sex and self-perception of health), the risk decrease to 63.9%. The novelty of these profiles is the knowledge that can be acquired and used by the health care professionals and educators to recognize the likelihood of falling, acting toward modifying potential risks and making falls preventable.

### Limitations and Strengths

This study has some limitations. One set of constraints relates to the small sample of participants for both the first evaluation (n = 186) and the second evaluation (n = 117). The reduced number of participants could be associated with the difficulty to identify participants who are living alone in a community context. In this particular cultural context, it is unusual to live alone; the stigma associated with living alone could potentially increase the challenges in recruiting. Because participants included in this study were living alone, the majority were afraid to “open” their houses and receive a “strange” person associated with the research team. Another point that needs to be taken into account is associated with our sample that was restricted to older people living in community dwellings with a small sample in northern Portugal; our sample cannot be generalized. Moreover, further studies should consider bigger samples to test this model. 

Despite the identified limitations, our results provide useful evidence for recognizing older people who have a low or high risk of falling, and the predictive model can better direct preventative efforts. On the one hand, there are several risk factors which we cannot modify, such as gender, age and, in some cases, where those people are living. On the other hand, many other risk factors exist that we can modify, and thus, modifications can be introduced to reduce the risk of falling. For example, we can add an exercise regime to improve balance or strength, adapt technical aids, rearrange therapeutic to minimize secondary effects or, from a social perspective, readjust the community resources. Another advantage of this study is related to the lack of models in a community context. This fact can be associated with the more controlled background, such as in a hospital or in nursing homes when compared with the community, because there are a vast number of factors that can promote falls. The findings showed that this model is valid and translational, allowing its implementation in clinical practice.

Another strength of this study is the possibility to identify different profiles, which allow developing our focus on different contexts, such as urban, considering the higher risk of falling and, to anticipate and personalize interventions based on modifiable risk factors (e.g., exercise). In short, this model can contribute to further research designs and potentiate future studies and new possibilities regarding the identification of risk scores to anticipate potential risks and prevent falls.

## 5. Conclusions

The predictive model developed revealed a satisfactory discriminatory performance of the model to identify the risk of falling among older people living alone in a community context. The results of this study have enabled the identification of various fall risk profiles in older people living alone. By being aware of these findings, we can facilitate more proactive interventions across multiple stakeholders in the social and health sectors. Furthermore, this knowledge helps enhance awareness of the modifiable risk factors that contribute to the likelihood of falls. Such involvement benefits not only the general population, but also specifically targets older individuals who live alone.

## Figures and Tables

**Figure 1 ijerph-20-06284-f001:**
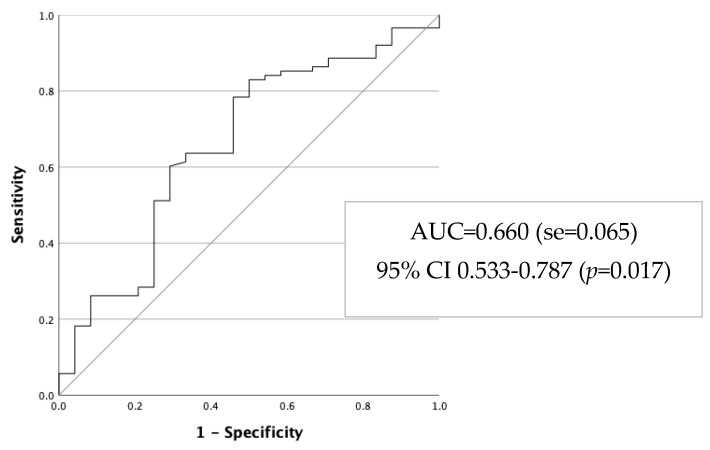
ROC curve.

**Table 1 ijerph-20-06284-t001:** Predictive model of risk of falling.

	B (se)	OR	CI 95%	*p*
Intercept	−0.33 (1.22)	0.717		0.786
Gender—female (ref: male.)	0.63 (0.52)	1.881	0.67–5.26	0.228
Age group (ref: 65–74 years)				0.600
75–84 years	0.48 (0.47)	1.609	0.64–4.07	0.315
85+ years	0.36 (0.94)	1.428	0.23–9.00	0.704
Zone—urban (ref: rural)	1.16 (0.52)	3.188	1.14–8.88	0.027
Social community resources—yes (ref: no)	0.42 (0.59)	1.521	0.48–4.83	0.477
Physical exercise—yes (ref: no)	−0.09 (0.51)	0.913	0.33–2.50	0.859
Self-perception of health (ref: very good + excelent)				0.410
Poor	−1.09 (1.14)	0.335	0.04–3.13	0.338
Reasonable	−1.21 (0.89)	0.298	0.05–1.71	0.174
Good	−1.56 (0.94)	0.209	0.03–1.31	0.095
Difficulty to keep standing—yes (ref: no)	0.51 (0.57)	1.659	0.54–5.07	0.374
Difficulty to sit and get up from the chair—yes (ref: no)	0.64 (0.65)	1.905	0.53–6.79	0.320
Difficulty to see—yes (ref: no)	0.21 (0.47)	1.231	0.49–3.09	0.658
Use of technical devices—yes (ref: no)	0.56 (0.79)	1.746	0.37–8.17	0.479
Hypertension—yes (ref: no)	0.71 (0.48)	2.035	0.80–5.19	0.137
Number of medications (ref: 0–3)				0.414
4–6	0.74 (0.55)	2.087	0.70–6.18	0.184
7+	0.38 (0.58)	1.457	0.46–4.59	0.520

B: coefficient; se: standard error; OR: odds ratio; CI: confidence interval.

**Table 2 ijerph-20-06284-t002:** Risk profiles.

	Profile 1	Profile 2	Profile 3
Gender	Female	Female	Female
Age group	85+	85+	85+
Zone	Urban	Urban	Rural
Social community resources	Yes	No	No
Physical exercise	No	Yes	Yes
Self-perception of health	Very-Good or Excellent	Very-Good or Excellent	Very-Good or Excellent
Difficulty to keep standing	Yes	No	No
Difficulty to sit and get up from the chair	Yes	No	No
Difficulty to see	Yes	No	No
Use of technical devices	Yes	No	No
Hypertension	Yes	Yes	No
Number of medications	7+	7+	0–3
Risk of falling	99.5%	94.4%	63.9%

## Data Availability

The data supporting this study’s findings are openly available on request from the authors.

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
