# Peer review of "Older People Living Alone: A Predictive Model of Fall Risk"

_ijerph, 2023, doi:10.3390/ijerph20136284_

Round 1

Reviewer 1 Report

This study presents with an interesting predictive model of fall risk. I would recommend authors to address a couple of issues.

English should be reviewed and edited, there are many minor mistakes. For instance, "persons" should be changed for "people" several times

Line 18, "predicte" models?

Line 20, is there a reason to conduct the analysis between two samples? You should detail what each sample encompasses

Line 39, "older persons" to "old people"

Line 46, "...and to rise" should be reviewed, this is not properly written

Line 48, "as a priority of one measure of policymakers" should be rewritten, I think it doesn't make any sense

Line 49. "Falls are multifactorial"? You should state the etiology, the reasons...but falls per se, are not multifactorial.

Line 52, I recommend to use "et al." instead

Line 69-70, "There is no effective instrument" really? There could not be a reliable one, but what do you mean with "effective"?

Line 73, "have the necessary skills and knowledge"

If you are conducting a test-retest reliability you are not really "validating", you are just checking the level of repeatability. Where is the validity analysis? There are many validity analyses you could've done. Internal consistency?

2.1 Aims. This is not only done with a test-retest, you should include other analyses

2.2. Informed consent should be stated independently, not as an inclusion crtieria

Line 109-110, could the assistance bias the results? You should be careful with this

Line 111. How did you blind the names? How could you know which first questionnaire belonged to which second?

Line 126. "Sociodemographic....outcomes?"

Line 161-162. So....you assessed fall frequency too? If so...why don't you state this under "outcomes"?

2.9. Validity. Actually, you could've tested yourself that validity. Your instrument is a combination of several questionnaires, it might not be as valid a those are.

Line 177. And what about sample 2?

Table 1. "p" should be in lower case, and a legend should be addedd below the table (what does "se" stand for?)

I appreciate the effort, but I recommend addressing this minor issues.

English should be improved considerably. Authors commit several times the same mistakes, and there are sentences that should be completely rewritten.

I understand English is not the first language of the authors, but English writing should be improved.

Author Response

We would like to thank you for your contributions that will enhance the quality of our manuscript titled “Older people living alone: a predictive model of fall risk”. The authors have made a careful English version. Each comment has been addressed and each adjustment made is highlighted in red in the manuscript.

Comments

This study presents with an interesting predictive model of fall risk. I would recommend authors to address a couple of issues.

English should be reviewed and edited, there are many minor mistakes. For instance, "persons" should be changed for "people" several times. Many thanks for the suggestions. The authors have made a careful English version.

Line 18, "predicte" models? Preditive models. Many thanks for the suggestions. The authors have made a careful English version.

Line 20, is there a reason to conduct the analysis between two samples? You should detail what each sample encompasses. The two sample are the same but evaluated at 2 different moments. With the first sample (moment 1), the predictive model was constructed. With the second sample (moment 2), the model was applied and evaluated.

Line 39, "older persons" to "old people". The authors have made a careful English version.

Line 46, "...and to rise" should be reviewed, this is not properly written. The authors have made a careful English version.

Line 48, "as a priority of one measure of policymakers" should be rewritten, I think it doesn't make any sense.  The authors have made a careful English version.

Line 49. "Falls are multifactorial"? You should state the etiology, the reasons...but falls per se, are not multifactorial. The sentence was removed.

Line 52, I recommend to use "et al." instead. The sentence was rewritten.

Line 69-70, "There is no effective instrument" really? There could not be a reliable one, but what do you mean with "effective"? The sentence was rewritten.

Line 73, "have the necessary skills and knowledge".  The authors have made a careful English version.

If you are conducting a test-retest reliability you are not really "validating", you are just checking the level of repeatability. Where is the validity analysis? There are many validity analyses you could've done. Internal consistency?  2.1 Aims. This is not only done with a test-retest, you should include other analyses. This work was intended to validate a model that assesses the risk of falling in older people living alone and not necessarily a measuring instrument. For this reason, we have kept this type of analysis.

2.2. Informed consent should be stated independently, not as an inclusion crtieria. Thank you for your suggestion. The sentence was rewritten.

Line 109-110, could the assistance bias the results? You should be careful with this - We thank you for your care in reviewing this item. The instruments were hetero-administered by a team of researchers trained for this purpose. The justification and option of the team is related to the difficulties in hearing, seeing and interpreting the questions of older people and, in this way - hetero administered - we intended to reduce bias.

Line 111. How did you blind the names? How could you know which first questionnaire belonged to which second?.  In order to maintain the anonymity of the participants, the team used an alphanumeric code.

Line 126. "Sociodemographic....outcomes?"The sentence was rewritten.

Line 161-162. So....you assessed fall frequency too? If so...why don't you state this under "outcomes"?. 2.9. Validity. Actually, you could've tested yourself that validity. Your instrument is a combination of several questionnaires, it might not be as valid a those are.  This work was intended to validate a model that assesses the risk of falling in older people living alone and not necessarily a measuring instrument. For this reason, we have kept this type of analysis.

Line 177. And what about sample 2? - The sentence was rewritten.

Table 1. "p" should be in lower case, and a legend should be added below the table (what does "se" stand for?) We added information to the footer.

I appreciate the effort, but I recommend addressing this minor issues. Thank you very much for your suggestions.

Reviewer 2 Report

The phrase "Evidence has shown that the world's population is getting older at an alarming rate" should be clarified with a bibliographic citation and, especially, with a focus on the gaps that occur at this level, according to different geographic regions.

The aim of this study was to build and validate a new model for predicting falls in elderly people living alone in the community. This was a test-retest reliability study to predict falls in elderly people living alone.

It was clarified that the study population consisted of people aged 65 years or over who lived at home in the city of Braga (District), and it was also confirmed that the project involved all elderly people who met the following eligibility criteria: ( a) live alone; (b) not having a cognitive deficit; (c) be able to communicate; (d) have given informed consent.

Where it reads "da Vida e da Saúde da Universidade do Minho (Reference number SECVS 033/2018)", the word "Saúde" and the word "of" must be separated, and the undue spacing between "SECVS" and "033/2018" as the text is not in justified format.

This study revealed a satisfactory discriminatory performance of the model to quantify the risk of falls in elderly people living alone in the community. Indeed, 12 variables obtained from data from the community context were introduced, which proved to be statistically relevant (bivariate analysis). The results showed that women fall more, although the authors point out that gender, as a risk factor for falls, is controversial.

It is perceived that elderly people living in urban community contexts have a higher risk of falling, because there are many more obstacles on the pavement and, therefore, they are exposed to a different risk every day.

In any case, there is the wish of the authors that the social resources of the community play a relevant role in helping people to minimize the risks, although, interestingly, the research results suggest that having social resources increases the risk of falling.

It was highlighted that maintaining an exercise activity appears to be a fall protector and may decrease the likelihood of having a recurrent fall. Indeed, the exercise programs that reduce the risk of falls mainly involve balance exercises and functional exercises, while the programs that probably effectively reduce falls include several categories of exercises, that is, their amplitude is more expressive.

It appears that the loss of strength and balance leads to various difficulties in managing activities of daily living, including self-care activities, with repercussions on this issue. In addition, the use of technical devices, most often associated with physical decline, is also a significant risk factor, confirmed by the results of the investigation that was undertaken. Pathologies also have a significant impact as a predictor of falls. For example, the authors mention that arterial hypertension is associated with the risk of falls.

Although the study has some limitations, such as the circumscribed case of the sample and the small number of participants (in the first evaluation (n=186) and in the second evaluation (n=117)), however, the study is useful and warns us to a significant problem.

The Discussion of Results justified this usefulness of the obtained data, but, in any case, the Conclusions are insufficient. Indeed, they only present this paragraph “The predictive model developed revealed a satisfactory discriminatory performance of the model to identify older people living alone in a community context with the risk of falling”, which, even taking into account what happened in the Discussion, deserved more schematization of the central ideas to be retained about the research carried out.

I suggest a practical layout of the conclusions, which highlights the strengths of the investigation.

Acceptable.

Author Response

Dear Reviewer 2

We would like to thank you for your contributions that will enhance the quality of our manuscript titled “Older people living alone: a predictive model of fall risk”. Each suggestion has been addressed and each adjustment made is highlighted in red in the manuscript.

The phrase "Evidence has shown that the world's population is getting older at an alarming rate" should be clarified with a bibliographic citation and, especially, with a focus on the gaps that occur at this level, according to different geographic regions. The authors introduced information about national context. 

The aim of this study was to build and validate a new model for predicting falls in elderly people living alone in the community. This was a test-retest reliability study to predict falls in elderly people living alone.  Thank you very much for your comment.

It was clarified that the study population consisted of people aged 65 years or over who lived at home in the city of Braga (District), and it was also confirmed that the project involved all elderly people who met the following eligibility criteria: ( a) live alone; (b) not having a cognitive deficit; (c) be able to communicate; (d) have given informed consent. Thank you very much for your comment. We removed the point d) in order to answer the reviewer 1.

Where it reads "da Vida e da Saúde da Universidade do Minho (Reference number SECVS 033/2018)", the word "Saúde" and the word "of" must be separated, and the undue spacing between "SECVS" and "033/2018" as the text is not in justified format. Thank you for your careful reading. The suggestions have been introduced.

This study revealed a satisfactory discriminatory performance of the model to quantify the risk of falls in elderly people living alone in the community. Indeed, 12 variables obtained from data from the community context were introduced, which proved to be statistically relevant (bivariate analysis). The results showed that women fall more, although the authors point out that gender, as a risk factor for falls, is controversial.Thank you for your careful reading. The suggestions have been introduced.

It is perceived that elderly people living in urban community contexts have a higher risk of falling, because there are many more obstacles on the pavement and, therefore, they are exposed to a different risk every day. Thank you for your careful and thorough reading of the results.

In any case, there is the wish of the authors that the social resources of the community play a relevant role in helping people to minimize the risks, although, interestingly, the research results suggest that having social resources increases the risk of falling. Thank you for your careful and thorough reading of the results.

It was highlighted that maintaining an exercise activity appears to be a fall protector and may decrease the likelihood of having a recurrent fall. Indeed, the exercise programs that reduce the risk of falls mainly involve balance exercises and functional exercises, while the programs that probably effectively reduce falls include several categories of exercises, that is, their amplitude is more expressive. Thank you for your careful and thorough reading of the results.

It appears that the loss of strength and balance leads to various difficulties in managing activities of daily living, including self-care activities, with repercussions on this issue. In addition, the use of technical devices, most often associated with physical decline, is also a significant risk factor, confirmed by the results of the investigation that was undertaken. Pathologies also have a significant impact as a predictor of falls. For example, the authors mention that arterial hypertension is associated with the risk of falls. Thank you for your careful and thorough reading of the results.

Although the study has some limitations, such as the circumscribed case of the sample and the small number of participants (in the first evaluation (n=186) and in the second evaluation (n=117)), however, the study is useful and warns us to a significant problem. Thank you for your careful and thorough reading of the results.

The Discussion of Results justified this usefulness of the obtained data, but, in any case, the Conclusions are insufficient. Indeed, they only present this paragraph “The predictive model developed revealed a satisfactory discriminatory performance of the model to identify older people living alone in a community context with the risk of falling”, which, even taking into account what happened in the Discussion, deserved more schematization of the central ideas to be retained about the research carried out. Thank you very much for the contributions on this point. The conclusion was reorganized and information on the expected implications for health and social policies was introduced, in response to the results obtained in this study.

Reviewer 3 Report

Falls in the elderly are a major medical and socio-community concern. Overall this paper is necessary and interesting, along with being well written. I only suggest minor changes:

-  In the introduction you could include the demographic background in Portugal(how many elderly people there are) and describe especially the possible about the change in household structures: increase of single-person households.

- Methodology: it would be interesting to include hypotheses that can then be discussed.

- Discussion: It would be convenient to briefly address the aspects of socio-health policies in Portugal and how this work could eventually impact, contribute, etc.

Author Response

Dear Reviewer 3

We would like to thank you for your contributions that will enhance the quality of our manuscript titled “Older people living alone: a predictive model of fall risk”. Each suggestion has been addressed and each adjustment made is highlighted in red in the manuscript.

Falls in the elderly are a major medical and socio-community concern. Overall this paper is necessary and interesting, along with being well written. I only suggest minor changes. Thank you for your suggestions introduced in this manuscript.

In the introduction you could include the demographic background in Portugal (how many elderly people there are) and describe especially the possible about the change in household structures: increase of single-person households. The authors introduced information about national context regarding number of older people in general and particularly the number os older people living alone.

Methodology: it would be interesting to include hypotheses that can then be discussed. Thank you very much for the suggestion. In order, not to substantially modify the content of the article, and going against the reviews of the other reviewers, we were unable to meet this point.

Discussion: It would be convenient to briefly address the aspects of socio-health policies in Portugal and how this work could eventually impact, contribute, etc. Thank you very much for the contributions on this point. The conclusion was reorganized and information on the expected implications for health and social policies was introduced, in response to the results obtained in this study.

Round 2

Reviewer 1 Report

No further comments to make.